# Data Augmentation Methods on Ultrasound Tongue Images for Articulation-to-Speech Synthesis

*Ibrahim Ibrahimov[1], Gábor Gosztolya[2], Tamás Gábor Csapó[3]*

[1]Institute of Informatics, University of Szeged, Szeged, Hungary
[2]MTA-SZTE Research Group on Artificial Intelligence, Szeged, Hungary
[3]Department of Telecommunications and Media Informatics, Budapest University of Technology and Economics, Budapest, Hungary

ibrahimkhaliloglu@gmail.com, ggabor@inf.u-szeged.hu, csapot@tmit.bme.hu

## Abstract

Articulation-to-Speech Synthesis (ATS) focuses on converting articulatory biosignal information into audible speech, nowadays mostly using DNNs, with a future target application of a Silent Speech Interface. Ultrasound Tongue Imaging (UTI) is an affordable and non-invasive technique that has become popular for collecting articulatory data. Data augmentation has been shown to improve the generalization ability of DNNs, e.g. to avoid overfitting, introduce variations into the existing dataset, or make the network more robust against various noise types on the input data. In this paper, we compare six different data augmentation methods on the UltraSuite-TaL corpus during UTI-based ATS using CNNs. Validation mean squared error is used to evaluate the performance of CNNs, while by the synthesized speech samples, the performace of direct ATS is measured using MCD and PESQ scores. Although we did not find large differences in the outcome of various data augmentation techniques, the results of this study suggest that while applying data augmentation techniques on UTI poses some challenges due to the unique nature of the data, it provides benefits in terms of enhancing the robustness of neural networks. In general, articulatory control might be beneficial in TTS as well.

**Index Terms**: data augmentation, silent speech interfaces, ultrasound tongue imaging, articulation-to-speech synthesis

## 1. Introduction

Speech production is a complex process that involves multiple parts working together to produce articulated sounds or words. However, in some cases, individuals may experience speech impairments due to damage to these components, such as the larynx (voice box) which can prevent them from producing audible speech. To address this issue, researchers have done studies on silent speech interface (SSI), which is an assistive technique to restore communication capabilities for those with speech impairments [1, 2, 3]. SSI targets the conversion of movements of articulatory parts (such as the tongue, lips, etc.) that are actively utilized in the process of speech production into hearable speech, a process known as articulation-to-speech synthesis (ATS). Historically, speech synthesis research focuses on text-to-speech synthesis (TTS), when the input is text or an estimated linguistic representation [4]. However, for ATS, very similar techniques can be applied that are already working for TTS, e.g. neural vocoders [5] or transformer networks [6]. Various techniques have been used to collect articulatory data as input for ATS, including electromagnetic articulography (EMA), surface electromyography (sEMG), permanent magnetic articulography (PMA), ultrasound tongue imaging (UTI), etc [7, 8, 9, 10, 11].

There exist two primary approaches for developing methods for speech synthesis from articulatory data in the domain of SSI [12]. The first method involves a process of recognition and synthesis, wherein first silent speech recognition (SSR) is utilized to obtain textual data from the articulatory input. Then this textual data serves as the input for a text-to-speech synthesis (TTS) system to produce a synthesized speech output [13, 14, 15, 16]. In contrast, the second approach, known as direct synthesis, involves transforming the articulatory input data into an intermediate representation (such as a mel-spectrogram) that can be utilized as input to a neural vocoder for generating synthesized speech [17, 18, 19, 20]. While the direct synthesis approach may not produce comparable speech in quality to TTS-generated speech, recent advancements in ATS have significantly improved the quality of the resulting speech output. For example, TaLNet can produce speech without intermediate spectral representation, using an encoder-decoder architecture [6]. As a summary, direct synthesis has become a viable option for use in SSI due to its ease of implementation and low latency which makes it more suitable for real-time application.

UTI has increasingly been utilized in SSI applications because it is a non-invasive and clinically safe method for tracking tongue movement [21, 22, 23]. Obtaining articulatory data for SSI typically requires more effort than collecting audio speech data [24]. Despite significant progress in the development of SSI, the additional challenges associated with collecting articulatory data have resulted in most studies relying on relatively small datasets in comparison to those used for speech recognition.

Convolutional neural networks (CNNs) have proven effective in the field of SSI using UTI data [25, 26, 27]. However, these models can be prone to overfitting and require relatively large amounts of data (e.g., at least several ten thousand data points) to achieve optimal performance. To address this issue, data augmentation techniques are often utilized to increase the quantity of training data available. In particular, Cao and his colleagues have applied data augmentation strategies to raw kinematic electromagnetic articulography signals in order to improve the performance of end-to-end SSR models on EMA input [28]. The various data augmentation methods performed differently, with a general accuracy increase in recognition performance, i.e. 5–20 % improvement measured using Phoneme Error Rate. An initial feasibility study tested four data augmentation strategies for UTI-based ATS [29], tested on an Azerbaijani ultrasound and speech dataset. Their results have shown that random scaling was the most effective approach, but the analysis was restricted to just validation MSE.

In this study, we test six distinct augmentation techniques directly to ultrasound images obtained from a freely available UTI dataset. The objective of this work is to enhance the ro-

bustness of the neural networks, and introduce variations to the existing dataset. The efficacy of each approach was evaluated by measuring the validation mean squared error of the trained CNN model, as well as the mel-cepstral distortion values [30] calculated to provide an objective measure of the quality of synthesized speech produced using the WaveGlow neural vocoder via direct synthesis. Additionally, the perceptual evaluation of speech quality (PESQ) scores [31] were obtained for 3 sentences per speaker from test set in this study. These metrics were used to assess the overall performance of the proposed methodology.

## 2. Dataset

In this research, we conducted experiments using four participants, two males (03mn, 04me) and two females (01fi, 02fe), selected from the UltraSuite-TaL80 database [32] (`https://ultrasuite.github.io/data/tal_corpus/`). This database was created by recording the tongue movements of speakers using an ultrasound system called "Micro" by Articulate Instruments Ltd. at a frame rate of 81.5 frames per second, in addition to recording their speech. Lip movements were also recorded, but were not utilized in this study. The audio and ultrasound data were synchronized using tools provided by Articulate Instruments Ltd. In our experiments, we only used ultrasound tongue images that had the 'aud' (audible read speech) and 'xaud' (shared audible read speech utterances) prompt tags. This allowed us to easily compare synthesized speech with the original speech. Considering this, the number of sentences read by each speaker was as follows: '01fi' read 180, '02fe' read 117, '03mn' read 169, and '04me' read 166 sentences. The data was then divided into training, validation, and test sets in a ratio of $80 - 10 - 10$.

## 3. Methods

### 3.1. Overview of ultrasound tongue image representations

Ultrasound images, at 64 x 842 pixels (as raw scanline representation, see e.g. Fig. 1), contain a large region of irrelevant information [21], i.e. some parts of the image are not directly related to the articulatory movement, like pixels above the palate. We resized the images to 64 x 128 pixels, as it has been shown that during UTI-based ATS, this does not cause significant information loss [33]. UTI images follow each other throughout the speech and create a sequence of images (a smaller-dimensional 'slice' of this 4D data is visualized in Fig. 2). The 3D figure shows that the center lines from the ultrasound images (see red box in Fig. 1 right) are visualized on the time-axis, with each time frame representing a duration of 12 ms. In this paper, for better understanding, a 2D representation, ultrasound 'kymograms' [34, 35] are used to visualize tongue image sequences and changes after each augmentation method. In the ultrasound tongue image sequence, raw ultrasound images come after each other, and in the creation of kymograms, from the middle part of each raw ultrasound image, one line of pixels are chosen to visualize the sequence (see again red box in Fig. 1 right). These kymograms show how the middle vertical line in the ultrasound image changes over time. In ultrasound kymogram (Fig. 3), the y-axis stands for distance from the starting point of the ultrasound wave till the end of it, which simply views the area between chin and upper hard palate, up to the ultrasound penetration depth, which is typically 80 or 90 mm with the above "Micro" system. Of course, such a kymogram is not the full

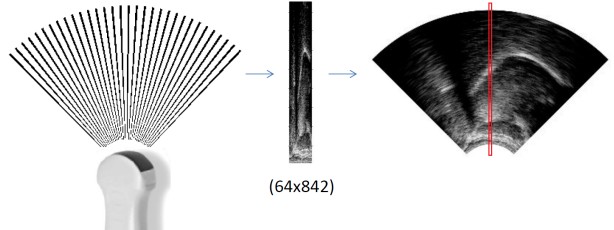

Figure 1: *Raw vs wedge-shaped ultrasound data. The red box indicates the center line of the 2D ultrasound image, used for kymogram representation (see Fig. 3).*

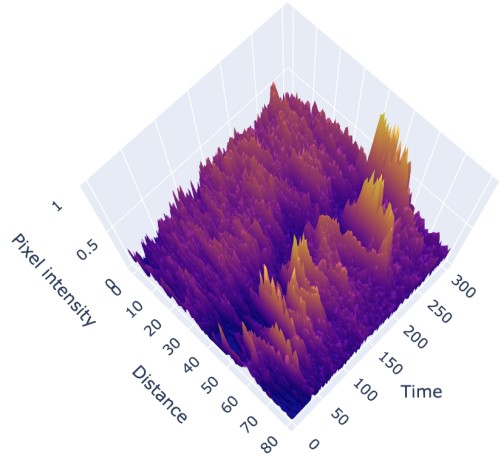

Figure 2: *Ultrasound tongue images sequence (3D figure).*

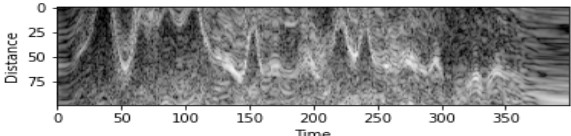

Figure 3: *Ultrasound tongue images sequence (kymogram).*

information that is available from the series of 2D ultrasound images, but they have been shown to be quite effective for visualizing general tendencies of articulatory movement [34, 35].

### 3.2. Data augmentation for articulation-to-speech synthesis

Data augmentation is a technique to simulate the process of imagination of neural networks. It is similar to how humans use their imaginations to understand and interpret the world around them [36]. Through different methods, variations of ultrasound tongue images can be created based on existing knowledge. These techniques improve the neural network's understanding of the ultrasound tongue image data and might help the generalization capability.

All the applied deep neural networks are constructed using two layers of two 2D convolutional + 2D Max Pooling layers, 'swish' activation function is used in every layers which have 30, 60, 90, 120 numbers of filter, kernel size of 13 and stride size 2, similarly to the previous study on UTI-based ATS [25]. During the training process, the neural network model was fed with the dataset for a total of 100 epochs and the training was

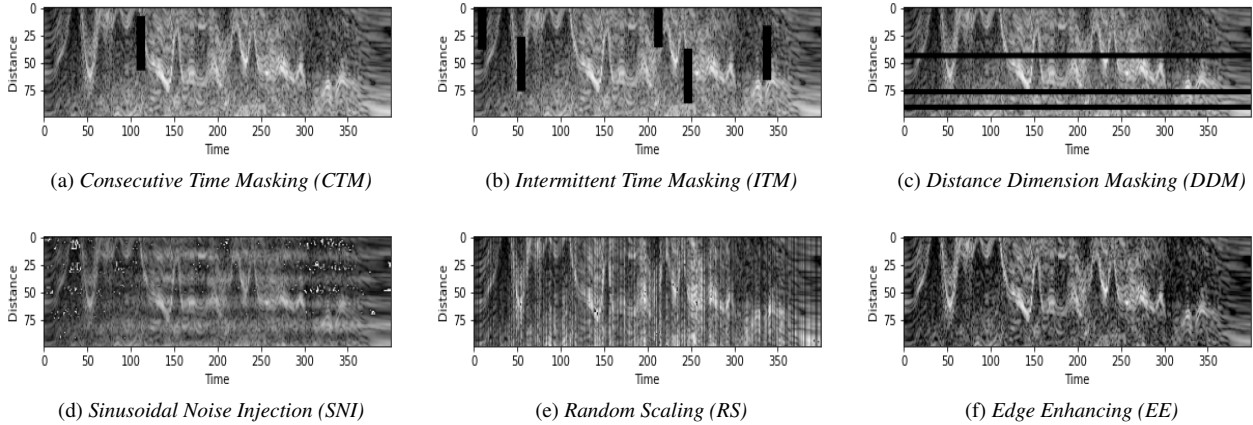

| (a) *Consecutive Time Masking (CTM)* | (b) *Intermittent Time Masking (ITM)* | (c) *Distance Dimension Masking (DDM)* |
| (d) *Sinusoidal Noise Injection (SNI)* | (e) *Random Scaling (RS)* | (f) *Edge Enhancing (EE)* |

Figure 4: *Sample UTI kymograms after data augmentation methods.*

conducted using a batch size of 128.

We set the augmentation ratio to 0.5 for all methods to prevent biased results by using 50% of augmented version of all UTI which is used in this study. By using a consistent augmentation ratio across all experiments, we can make valid comparisons between the baseline and augmented results without introducing any confounding variables. It is worth noting that the specific parameters for each method were determined on validation set through experimentation based on the results.

### 3.2.1. Baseline

To ensure a fair comparison, we expanded the dataset by including 50% of the same ultrasound images for each speaker. This was done to have the same amount of data as when using augmentation methods. This allowed us to observe the differences between using augmentation methods and using the same amount as the augmentation methods but with original data without any modification. The aim is to emphasize the benefits provided by data augmentation methods on ultrasound tongue images.

### 3.2.2. Consecutive time masking (CTM)

Consecutive time masking (CTM) is used to enhance the learning process of the network by preventing it from seeing certain parts of the ultrasound images, similarly to [37]. CTM involves consecutively masking 10 frames along the time-axis. Over time, the starting point for masking is randomly selected, while the starting point for the distance-axis is randomly chosen from a range between 50 and 150. This is because our raw ultrasound images are resized to 64 x 128 pixels, with the second value representing distance. The masked frames have a length of 50 along the distance-axis. In Fig. 4a, the effect of the CTM augmentation technique is visualized as an ultrasound kymogram representation.

### 3.2.3. Intermittent time masking (ITM)

Next, we utilized the intermittent time masking (ITM) method, which is similar to CTM. However, instead of consecutive masking along the time-axis, frames are masked intermittently with random starting points. Specifically, 5 sections are masked intermittently, with each having the capacity to mask 10 consecutive frames, and a length of 50 which has a starting point

randomly chosen between 50 and 150 across the distance-axis. While some masked sections may be consecutive due to the random starting points, there is no overlap between them. The visual representation of ITM on ultrasound tongue images are depicted in Fig. 4b.

### 3.2.4. Distance Dimension Masking (DDM)

In addition to time masking techniques, another type of masking what we term the distance dimension masking (DDM) method is utilized in this work. DDM involves masking sections of ultrasound tongue images in the distance-axis which was inspired by previous work by Cao and colleagues [28], who investigated the use of articulatory dimension reduction as a form of data augmentation. In DDM, 3 intermittent sections consisting of 5 consecutive lines from the distance-axis are masked. The purpose of this method is to encourage the network to learn more robust features that are invariant to changes in distance dimensions. By masking out certain sections of the image, the network is forced to focus on other parts of the image, which can help improve its ability to recognize features that are relevant to speech production. The visualization as an ultrasound kymogram provides a clear illustration of the DDM method in Fig. 4c.

### 3.2.5. Sinusoidal noise injection (SNI)

Sinusoidal noise injection (SNI) is a useful method for understanding cyclic patterns in data. This technique involves adding sinusoidal waves as noise to the input data as shown in Figure 4d. The amplitude of the sinusoidal waves was calculatd as the average value of pixel intensity values across time-axis multiplied by a scaling factor. The scaling factor in our study was set to 0.02, and the number of oscillations per second (Hz) was 40. The sinusoidal noise added would be represented by $I(t) = 0.02 \cdot M \cdot sin(2\pi \cdot 40t)$, where $M$ is the mean pixel intensity.

### 3.2.6. Random scaling (RS)

The random scaling (RS) data augmentation method is a technique that involves the random modification of pixel intensity values along the time-axis of an ultrasound tongue image (Fig. 2). The scaling factor, which is a random value chosen from the range $[0.8 : 1.4]$, is the key parameter for this method. In practice, if the scaling factor is less than one, the intensity

of the pixel will be reduced, whereas applying a scaling factor greater than one will lead to an increase in the pixel intensity. The visual result of randomly scaling the pixel intensity is illustrated in Fig. 4e.

### 3.2.7. Edge Enhancing (EE)

We also explored the use of edge enhancing (EE), which creates subtle bright and dark contrasts on either side of edges present in the image, thereby enhancing their visibility and making them appear more distinct. This technique might be particularly suitable for ultrasound tongue images due to the presence of bright regions in the image that correspond to the tongue contour. By enhancing the edges in these regions, the edge enhancement method can aid in better depiction of the tongue contour and improve the overall quality of the ultrasound image (Fig. 4f). We applied a Gaussian blur filter to the ultrasound image with a kernel size of 15, then the blurred image is combined with the original image. This combination is achieved by multiplying the original image by a weight of 1.5, and the blurred image by a weight of -0.5, then adding the two images together.

### 3.3. WaveGlow neural vocoder

Similarly to the original WaveGlow paper [38], 80 bins were used for mel-spectrogram using librosa mel-filter defaults. FFT size and window size were both 1024 samples. For hop size, we chose 270 samples, in order to be in synchrony with the articulatory data. This 80-dimensional mel-spectrogram served as the training target of the neural network. NVIDIA provided a pretrained WaveGlow model using the LJSpeech database (WaveGlow-EN) [39]. In the synthesis phase, an interpolation in time was necessary, as the original WaveGlow models were trained with 22 kHz speech and 256 samples frame shift; for this we applied bicubic interpolation. Next, to smooth the predicted data, we used a Savitzky-Golay filter with a window size of five, and cubic interpolation, similarly to [25]. Finally, the synthesized speech is the result of the inference with the trained WaveGlow model (EN) conditioned on the mel-spectrogram input [38].

## 4. Results and Discussion

### 4.1. Objective evaluation

To objectively evaluate the effectiveness of various augmentation methods on ultrasound tongue images, we first employed the mean squared error metric on the validation set to differentiate between the results obtained from the augmentation methods and the baseline approach for each speaker individually (Table 1). We observed that the utilization of augmentation methods yielded diverse outcomes for each speaker when compared to the baseline approach. A decreased V-MSE value is indicative of enhanced learning by the neural network using specific augmentation methods. Namely, for speakers '01fi' and '04mn', the ITM method, for speaker '02fe', the EE method, for speaker '03mn', the DDM method yielded the lowest results. Nevertheless, the mean values obtained from the V-MSE metric demonstrate that there were only slight improvements attained. Notably, the DDM technique provided the optimal score among all the methods, including the baseline approach.

After using WaveGlow neural vocoder to synthesize speech for each speaker, we calculated MCD values [30] as an objective metric (results in Table 2). Lower MCD values indicate higher spectral similarity (note that the values are in the range of 2.0

Table 1: *Validation Mean Squared Error (V-MSE) results (lower is better, bold is the best system).*

| Speaker | Baseline | CTM | ITM | DDM | SNI | RS | EE |
|---|---|---|---|---|---|---|---|
| 01fi | 0.212 | 0.197 | **0.189** | 0.194 | 0.200 | 0.198 | 0.202 |
| 02fe | 0.320 | 0.300 | 0.325 | 0.301 | 0.293 | 0.290 | **0.282** |
| 03mn | 0.168 | 0.180 | 0.176 | **0.114** | 0.156 | 0.157 | 0.152 |
| 04me | 0.188 | 0.183 | **0.181** | 0.205 | 0.182 | 0.186 | 0.183 |
| Mean | 0.222 | 0.215 | 0.218 | **0.203** | 0.208 | 0.207 | 0.204 |

Table 2: *Mel-cesptral distortion (MCD) results (higher is better, bold is the best system).*

| Speaker | Baseline | CTM | ITM | DDM | SNI | RS | EE |
|---|---|---|---|---|---|---|---|
| 01fi | **2.175** | 2.040 | 2.078 | 2.059 | 2.148 | 2.103 | 2.075 |
| 02fe | **2.038** | 1.746 | 1.952 | 1.907 | 1.936 | 1.930 | 1.911 |
| 03mn | 2.218 | 2.178 | **2.417** | 1.978 | 2.117 | 2.159 | 2.115 |
| 04me | 2.162 | 2.111 | **2.197** | 1.996 | 2.069 | 2.135 | 2.065 |
| Mean | 2.148 | 2.018 | **2.161** | 1.985 | 2.067 | 2.081 | 2.041 |

which is not typical for standard text-to-speech, but we were using a custom MCD implementation). In our experiments, all augmentation techniques, except for the ITM method applied to speakers '03mn' and '04me', achieved lower MCD values compared to the baseline approach. Notably, for speaker '01fi', the RS method, for speaker '02fe', the CTM method, and for speakers '03mn' and '04me', the DDM method demonstrated the best performance in terms of achieving the lowest MCD values among the proposed augmentation techniques. According to the mean MCD values, similarly to V-MSE, the DDM approach illustrated the greatest improvement on ultrasound images, as evidenced by the most significant decrease compared to the baseline approach.

In addition to evaluating the MCD values, we further assessed the quality of synthesized speech by calculating the PESQ scores [31] for 3 individual sentences of each speaker from the test set. The PESQ score is a widely-used objective metric that is based on a model of human auditory perception and provides a measure of the similarity between the synthesized and original speech. A higher PESQ score indicates better speech quality. Our results indicate that, for speaker '02fe', most of the augmentation techniques resulted in an improvement in the quality of synthesized speech, except for sentence '084' where the ITM method led to a reduction in the PESQ score. For other sentences from each speaker, we observed that different augmentation methods led to higher PESQ scores. When evaluating the quality of synthesized speech for speaker '04mn' and sentence '082', it was surprising to find that the baseline approach resulted in the highest PESQ score and thus produced the best speech quality. However, based on the average PESQ scores, the SNI technique illustrated the highest score among all the methods evaluated (Table 3).

While the proposed data augmentation techniques in this study are effective for RGB images and EMA signals, their application to ultrasound tongue images is more challenging due to the larger dimension (compared to EMA) and different format of the data (compared to RGB images). The findings of this study suggest that while applying data augmentation techniques on UTI poses some challenges due to the unique nature of the data, it provides benefits in terms of enhancing the robustness of neural networks. In the context of the current study, for example, Cao et al. found that consecutive time masking was the most effective for EMA-based SSR [28], whereas the initial study of

Table 3: *Perceptual Evaluation of Speech Quality (PESQ) scores of three sentences per speaker from test set (higher is better, bold is the best system).*

| Sentence | Baseline | CTM | ITM | DDM | SNI | RS | EE |
|---|---|---|---|---|---|---|---|
| '01fi' - 007 | 1.133 | **1.156** | 1.148 | 1.152 | 1.141 | 1.133 | 1.151 |
| '01fi' - 018 | 1.099 | 1.100 | 1.115 | **1.116** | 1.102 | 1.097 | 1.107 |
| '01fi' - 131 | 1.239 | **1.273** | 1.269 | 1.248 | 1.228 | 1.223 | 1.257 |
| '02fe' - 066 | 1.217 | 1.327 | 1.309 | 1.334 | **1.395** | 1.373 | 1.373 |
| '02fe' - 082 | 1.239 | 1.328 | 1.272 | 1.324 | 1.307 | 1.333 | **1.453** |
| '02fe' - 084 | 1.262 | 1.370 | 1.247 | 1.337 | **1.398** | 1.369 | 1.334 |
| '03mn' - 007 | 1.324 | 1.290 | 1.350 | **1.429** | 1.358 | 1.340 | 1.364 |
| '03mn' - 018 | 1.341 | 1.355 | 1.373 | **1.508** | 1.369 | 1.330 | 1.360 |
| '03mn' - 131 | 1.398 | 1.471 | 1.446 | 1.233 | **1.525** | 1.456 | 1.514 |
| '04me' - 066 | 1.236 | 1.309 | **1.371** | 1.310 | 1.287 | 1.217 | 1.348 |
| '04me' - 082 | **1.620** | 1.509 | 1.533 | 1.490 | 1.512 | 1.515 | 1.520 |
| '04me' - 084 | 1.520 | 1.468 | 1.559 | 1.463 | 1.530 | 1.490 | **1.577** |
| Mean | 1.303 | 1.338 | 1.339 | 1.335 | 1.341 | 1.336 | **1.346** |

Ibrahimov and his colleagues on UTI-based ATS concluded that random scaling was most useful [29]. The present research extended the above findings on a larger dataset (4 speakers of the UltraSuite-TaL database as opposed to a single speaker in [29]) and with two more data augmentation methods, followed by a more thorough analysis of the results, including MSE and PESQ measures.

### 4.2. Subjective evaluation

By informally listening to the synthesized samples, one can hear slight differences as a result of data augmentation. However, the authors felt that such tiny details would not be observable for the average human ear, so we decided not to do a full subjective evaluation, and our results are solely based on the objective evaluation. Synthesized samples are at `http://smartlab.tmit.bme.hu/ssw12-UTI-augmentation`.

## 5. Conclusions and Future work

This study highlighted several different data augmentation methods on UTI for ATS, inspired by a similar work on EMA data [28] and extending the initial study of [29]. The findings suggest that the effectiveness of data augmentation techniques may vary depending on individual speaker characteristics, and therefore, a specifically chosen augmentation approach may be necessary to achieve optimal performance for each individual case. The results suggest that by carefully selecting different augmentation methods on ultrasound tongue images, it is possible to enhance the performance of neural networks (in terms of objective measures), but it is a question whether such improvement of performance causes audible quality improvement of synthesized speech in articulation-to-speech synthesis. Using various data augmentation techniques leads to increased robustness of the neural network when trained on the UTI dataset.

In the general context of text-to-speech synthesis, articulatory information has been shown to be effective in improving the performance of HMM-based and DNN-based TTS – in an overview, Richmond and his colleagues summarize the use of articulatory data in speech synthesis applications [24]. Besides, there have been examples where articulatory control was shown to be beneficial in TTS, including [40], [11], or [41]; therefore, we expect that the above ATS experiments might be useful for future use in TTS as well.

In future work, subsequent investigations are planned to extend the application of the evaluated data augmentation techniques to other modalities of articulatory data acquisition (e.g. vocal tract MRI or lip video), thus expanding our understanding of their potential benefits in improving ATS for SSI. It is also a future plan to apply automatic hyperparameter optimization, and compare the proposed mechanisms with a heavy dropout during training, which might have a similar regularization effect.

## 6. Acknowledgment

The research was funded by the National Research, Development and Innovation Office of Hungary (FK 142163 grant). Furthermore, this work was supported by the European Union project RRF-2.3.1-21-2022-00004 within the framework of the Artificial Intelligence National Laboratory and project TKP2021-NVA-09, implemented with the support provided by the Ministry of Innovation and Technology of Hungary from the National Research, Development, and Innovation Fund, financed under the TKP2021-NVA funding scheme. T. G.Cs.'s research was supported by the Bolyai János Research Fellowship of the Hungarian Academy of Sciences, and by the ÚNKP-22-5-BME-316 New National Excellence Program of the Ministry for Culture and Innovation from the source of the National Research, Development and Innovation Fund. The Titan X GPU used was donated by NVIDIA. We would like to thank the MTA-ELTE Lingual Articulation Research Group for providing the equipment necessary for the articulatory recordings and Beiming Cao and his colleagues at the University of Texas at Austin, USA for the excellent research idea [28].

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
