# OpenReview forum: "Data Augmentation Methods on Ultrasound Tongue Images for Articulation-to-Speech Synthesis"
_Interspeech.org/2023/Workshop/SSW — SSW12_

### Official Review · Reviewer_QRJg · 2023-06-05
**Minor improvements using various data augmentation techniques**

**Rating:** 6
**Confidence:** 3

**Review:**

This paper discusses various ways to augment tongue ultrasound data for training an articulation to speech synthesizer. Authors did not perform a subjective evaluation because the differences between different versions are hardly audible. They do present several objective measures and show that the various methods did not lead to large differences in the outcome.

---

### Official Review · Reviewer_zve1 · 2023-06-13
**This paper presents an objective analysis of the impact of different data augmentation techniques for ultrasound tongue imaging applied to the articulation-to-speech (ATS) synthesis. The augmentation approaches are mainly focused on 'data augmentation through error induction', as the aim of the authors is to look into human knowledge-driven approaches. Results highlight that through data augmentation it is possible to marginally improve objective metrics but with no perceptual impact.**

**Rating:** 6
**Confidence:** 4

**Review:**

Strengths:

Very good quality introduction, and educative to the point that as a non-expert in articulatory synthesis I feel I have a better grasp of the problem. Overall very high quality presentation, writing and flow of the paper. Good job on the writing.

Comprehensive analysis of possible data augmentation techniques based on the literature.

Improvement areas:

(minor) literature on TTS is very sparse (e.g. picking only 'transformers' as a reference?). This is not critical per se, so definitely a minor comment.

(minor) I would have liked a bit more background on the problems around ATS, less tailored to the specific application.

I am missing an ablation study of the impact of amount of data in data augmentation, and somewhat an understanding of how the quality of the augmented data helps drive better results or not. This overall limits the novelty of the work-

A bit disappointing to only see objective metrics on synthesis quality. Would have been very interesting to see how they relate with human listening annotations, even simple ones to get an understanding of what is the overall standing of the ATS problem in terms of quality. Although I understand the justification from the authors, it is weird to provide objective metrics while at the same time mentioning that perceptually they are irrelevant! This is single-handedly removing weight to the claim that 'The results suggest that by carefully selecting different augmentation methods on ultrasound tongue images, it is possible to enhance the performance of neural networks and slightly improve the quality of synthesized speech in articulation-to-speech synthesis'.

I would encourage to add some bolding/highlighting on tables 1, 2 and 3 to make the information easier to digest. Even adding in the caption 'higher/lower is better, bold is the best system' would lead to a cleaner and faster information visualization.

Comments/questions

Considering you have a mostly objective-metrics driven system, did you consider hyper-parameter optimization for your tunings instead of manual corrections? Given the explanation in section 3.2 it feels somewhat manually-driven which somewhat lowers reproducibility of the results (and risks sub-optimal exploration)
Considering how the data augmentation mechanisms are mostly around 'obstruction' of the information in the signal. How would the proposed mechanisms compare with a heavy dropout during training?

---

### Decision · Program_Chairs · 2023-06-15

**Decision:**

Accept

**Comment:**

SSW2003 received 45 papers. The acceptance rate is 82%. We are pleased to inform you that your paper has been accepted by the SSW2023 Program Committee. Please read the reviews carefully and submit your camera-ready paper by June 28th. Most reviewers performed a detailed review. Please answer to their questions and consider their comments. Note that camera-ready papers are credited with one extra page to allow authors to consider reviewers’ suggestions. So max 7 pages in total including figures & refs.
The deadline for submitting the revised version (with full non-anonymized authors and refs!) is 28th June.